evolution/taxonomy and systematics/ palaeontology

Echinodermata, ophiuroids, phylogeny, evolution, deep sea

**Author for correspondence:**
Ben Thuy
e-mail: bthuy@mnhn.lu

# New fossils of Jurassic ophiurid brittle stars (Ophiuroidea; Ophiurida) provide evidence for early clade evolution in the deep sea

Ben Thuy[1], Lea D. Numberger-Thuy[1] and Tania Pineda-Enríquez[2]

[1]Department of Paleontology, Natural History Museum Luxembourg, 25, rue Münster, 2160 Luxembourg City, Luxembourg
[2]Department of Biology, Division of Invertebrate Zoology, Florida Museum of Natural History, University of Florida, 1659 Dickinson Hall, Gainesville, FL 32611, USA

BT, 0000-0001-8231-9565; LDN-T, 0000-0001-6097-995X

Understanding of the evolutionary history of the ophiuroids, or brittle stars, is hampered by a patchy knowledge of the fossil record. Especially, the stem members of the living clades are poorly known, resulting in blurry concepts of the early clade evolution and imprecise estimates of divergence ages. Here, we describe new ophiuroid fossil from the Lower Jurassic of France, Luxembourg and Austria and introduce the new taxa *Ophiogojira labadiei* gen. et sp. nov. from lower Pliensbachian shallow sublittoral deposits, *Ophiogojira andreui* gen. et sp. nov. from lower Toarcian shallow sublittoral deposits and *Ophioduplantiera noctiluca* gen. et sp. nov. from late Sinemurian to lower Pliensbachian bathyal deposits. A Bayesian morphological phylogenetic analysis shows that *Ophiogojira* holds a basal position within the order Ophiurida, whereas *Ophioduplantiera* has a more crownward position within the ophiurid family Ophiuridae. The position of *Ophioduplantiera* in the evolutionary tree suggests that family-level divergences within the Ophiurida must have occurred before the late Sinemurian, and that ancient slope environments played an important role in fostering early clade evolution.

# 1. Introduction

Earth's ecosystems are so complex and diverse that their scientific assessment is generally done using model organisms as a manageable subset representative for the whole. One of the groups that recently emerged as a promising model organism to explore marine biodiversity and evolution are the ophiuroids, the slender-armed cousins of the starfish. Present-day distribution data combined with molecular and morphological phylogenies calibrated against the fossil record have yielded significant insights to better understand how marine biodiversity has evolved (e.g. [1–4]). The backbone of such studies is a comprehensive evolutionary tree of the model group.

In ophiuroids, phylogenetic estimates have recently progressed to the point that the molecular data available are not only among the most exhaustive in marine invertebrates [5] but furthermore agree with morphological estimates [6]. Despite these advances, the deep divergences are still poorly understood. The Ophiuroidea are an ancient class, with first representatives emerging in the Ordovician [7] and crown-group diversification starting in the late Paleozoic [8]. The fragmentary knowledge of the fossil record, however, hampers the use of the Ophiuroidea as a model organism to explore deep-time biodiversity patterns.

The major living ophiuroid clades originated during the early Mesozoic [8], often via extinct stem members with intermediate morphologies [9]. This implies that the basal diversification of the crown-group ophiuroids can only be fully understood using fossils of the extinct representatives [6]. A significant share of these ancient clades, however, is assumed to have originated in the deep sea [1,3]. Considering the notoriously poor fossil record of deep-sea biota (e.g. [10–11]), basal representatives of the ancient living ophiuroid clades are difficult to identify in the fossil record.

Here, we explore the early evolutionary history of the order Ophiurida, assumed to have started in the Late Triassic to Early Jurassic [5]. We analyse the phylogenetic position of previously known Jurassic members of the Ophiurida and describe three new ophiuroid fossils from Lower Jurassic shelf and shallow bathyal deposits. We show that the new fossils are part of the ancient stock of the living ophiurid families Ophiuridae and Ophiopyrgidae. The two families are among the most speciose and widespread in the present-day oceans, both in shallow and deep-water environments. Whereas molecular evidence suggests a mid-Cretaceous divergence of the Ophiuridae and Ophiopyrgidae [5], our new fossil data provide evidence that their evolutionary history already began in the Early Jurassic. Our results furthermore suggest that the currently known pre-Cenozoic fossil record of the two families is sparse (e.g. [12]) because a significant part of their early evolution took place at poorly sampled bathyal palaeo-depths.

# 2. Material and methods

## 2.1. Fossil specimens and geological background

All fossils described herein were retrieved from the sieving residues of bulk sediment samples, screen-washed with tap water, air-dried and sorted under a dissecting microscope using a small paintbrush. The fossil material comprises three taxa from three different localities.

(1) Articulated skeleton fragments and dissociated skeletal plates described below as *Ophiogojira labadiei* gen. et sp. nov. were collected from a succession of clays and thin lumachellic beds with siderite nodules, exposed near Sedan, Ardennes, France (figure 1), dated to the late Early Pliensbachian (Davoei chronozone, Early Jurassic), and deposited in a shallow, sublittoral, near-coastal setting. A detailed description of the locality, including its stratigraphy, depositional setting and echinoderm fossil content was provided by Thuy *et al.* [13].

(2) Dissociated lateral arm plates described below as *Ophioduplantiera noctiluca* gen. et sp. nov. were picked from the sieving residues of marls from the Kehlbach and Scheck Members within the Adnet Formation, exposed in the Glasenbach Gorge near Salzburg, Austria (figure 1), dated to the late Sinemurian to late Pliensbachian (Raricostatum to Margaritatus chronozones, Early Jurassic), and deposited at bathyal palaeo-depths. Further details on the locality and the depositional setting can be found in Thuy *et al.* [11] and Honigstein *et al.* [14].

(3) Dissociated lateral arm plates described below as *Ophiogojira andreui* gen. et sp. nov. were picked from the sieving residues of dark grey marls temporarily exposed during construction works at the site Uerschterhaff near Sanem, Luxembourg (figure 1). Lithostratigraphically, the marls

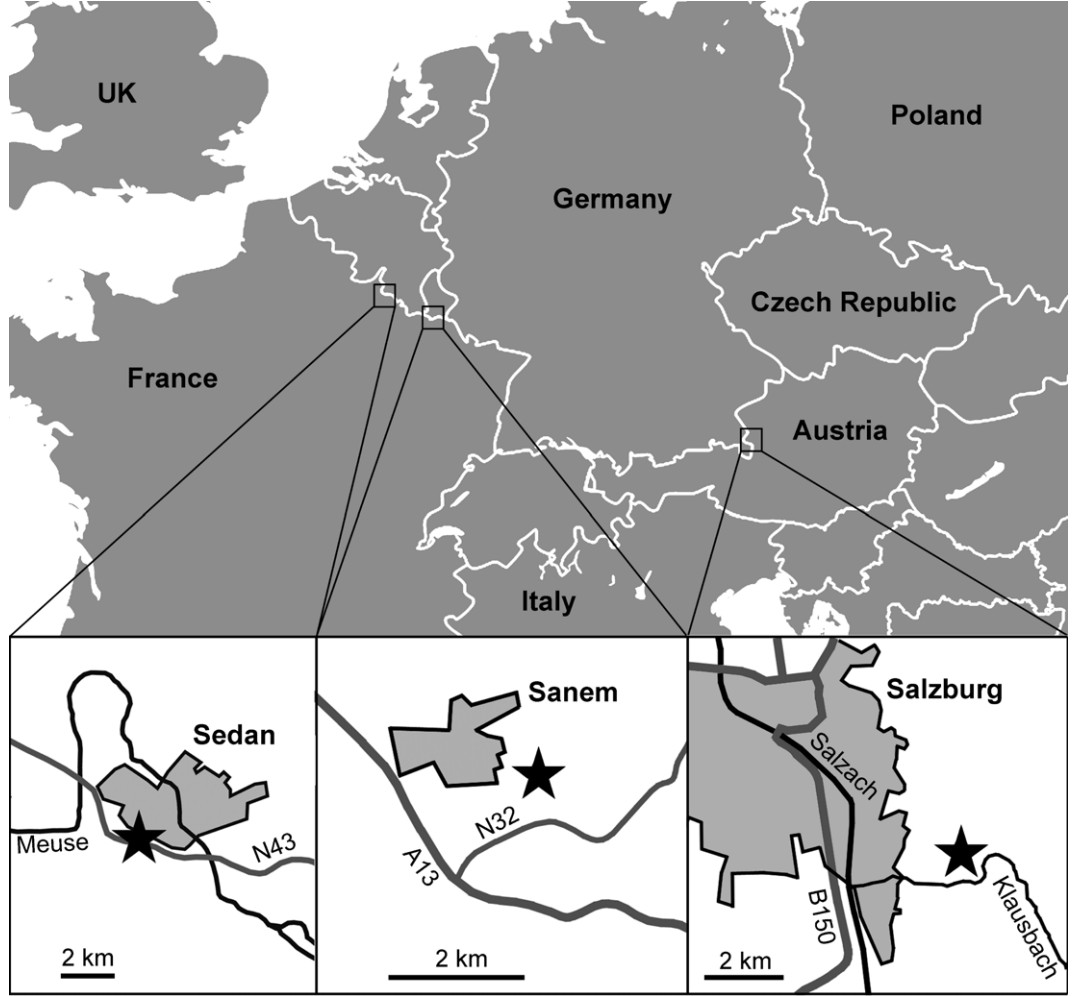

**Figure 1.** Map showing the position of the localities that yielded the fossil described herein, each indicated by a star.

correspond to the Ottempt Member of the Aubange Formation [15]. Their earliest Toarcian age (Tenuicostatum chronozone, Early Jurassic) is confirmed by the co-occurrence of dactylioceratid ammonites and the belemnite *Passaloteuthis laevigata* (Zieten, 1831) [16–19].

All three localities were located in the western Tethys Ocean during Early Jurassic times (figure 2). Sedan and Sanem were both on the southern shores of the emerged London-Brabant massif amid the shallow epicontinental northwestern Peritethys. The Glasenbach Gorge was on the southern slope of a deep subbasin of the oceanic Neotethys northern branch [20].

Selected specimens from all three localities were cleaned in an ultrasonic bath, mounted on aluminium stubs and gold-coated for scanning electron microscopy using a JEOL Neoscope JCM-5000. Type and other figured specimens were deposited in the palaeontological collection of the Natural History Museum Luxembourg (acronym MnhnL OPH).

Morphological terminology follows Thuy & Stöhr [6,21], Stöhr *et al.* [7] and Hendler [22]. We adopt the classification by O'Hara *et al.* [5,23].

## 2.2. Phylogeny

In order to explore the phylogenetic position of the fossils described herein within the living Ophiuroidea and with respect to previously published fossils with assumed ophiurid affinities, we performed a Bayesian inference analysis using the matrix elaborated by Thuy & Stöhr [6] and modified by Thuy & Stöhr [9] as a basis. Because our analysis focused on the Ophiurida, we expanded taxon sampling within this order to capture the morphological spectrum of its living representatives more exhaustively (table 1). The fossil taxa added to the matrix comprise the type species of their respective genera each described herein: *Ophioduplantiera noctiluca* gen. et sp. nov. and *Ophiogojira labadiei* gen. et

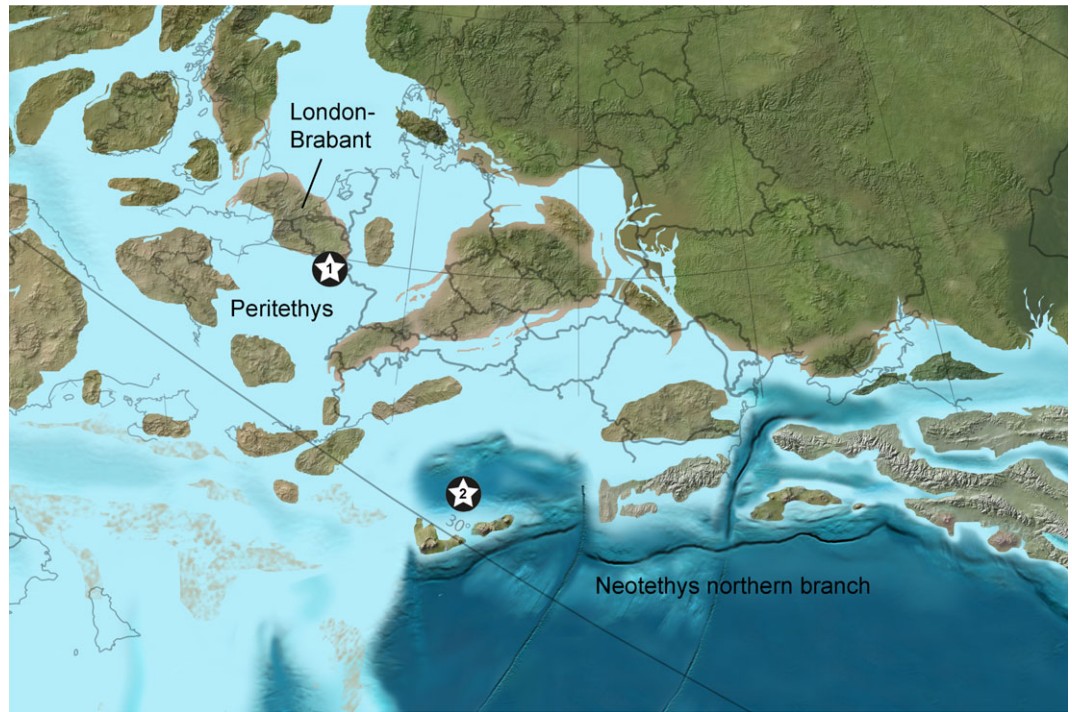

**Figure 2.** Palaeogeographic map of Europe during the Early Jurassic showing the position of the Sedan and Sanem localities (star no. 1) and the Glasenbach Gorge locality (star no. 2).

sp. nov. *Ophiogojira andreui* gen. et sp. nov. is not included because it is largely similar to *O. labadiei* but only known from dissociated lateral arm plates, thus adding no extra value to the phylogeny. In addition to the new fossils, we included several previously published ophiurid fossil taxa (table 1), taking into account the few known pre-Cenozoic representatives of the suborder Ophiurina (Ophiuridae and Ophiopyrgidae) and the most exhaustively known Jurassic members of the suborder Ophiomusina (Ophiomusaidae and Ophiosphalmidae). The final matrix comprises 63 ingroup taxa (of which 13 are fossils), and *Aganaster gregarius* (Meek & Worthen, 1869) as outgroup taxon.

Character scoring was performed as outlined by Thuy & Stöhr [6,9] using the same list of characters (and character acronyms), with the following exceptions as a result of the ongoing improvement of our ophiuroid character matrix (see electronic supplementary material). Characters of the distal edge of the abradial genital plate and associated papillae were modified to better describe the morphological spectrum within the Ophiurida: GP-8 deleted; GP-7 abradial genital plate with papillae extending on latero-distal edge of disc: no (0), yes (1); GP-9 shape of distal genital papillae: spine-like (0), block-like (1). Similarly, characters of the tentacle openings were adapted to the now increased morphological spectrum of the ophiurid clades: LAP-TP-1 tentacle opening in proximal and/or median segments outside disc: developed as ventralwards or ventro-distalwards pointing between-plate notch (0), modified into within-plate perforation or distalwards pointing cut-out (1); LAP-TP-2 tentacle opening of median segments pointing: ventralwards to ventro-distalwards (0), distalwards (1). The expansion of the Ophiomusina sampling prompted us to include a new character of the dorsal edges of the lateral arm plates: LAP-G-6 dorsal edge distalwards ascending: no (0); yes (1). Finally, Hendler's [22] re-assessment of the ophiuroid buccal skeleton required a revision of the relevant characters in our matrix as follows: M-SP-1 second tentacle pore position: deep in mouth slit and vertical (0), shallow and oblique (1); M-SP-3 tentacle scale(s) at the first ventral arm plate: absent (0), present (1); M-Pa-T-2 tooth papillae: absent (0), two to several (1), forming a cluster (2), M-Pa-T-4 MP2 position: deleted; M-Pa-T-3 buccal scale: absent (0), present (1); M-Pa-T-1 oral papillae (*sensu* lato) number: single row along the jaw edge (0), multiple rows covering jaws (1), none (2); M-Pa-T-7 oral papillae (*sensu* lato) shape: block-shaped (0), rounded (1), paddle-shaped (2), spiniform (3); LAP-I-7: large dorsal contact surface with opposite LAP: absent (0), present (1); LAP-I-8: large ventral contact surface with opposite LAP: absent (0), present (1).

Bayesian inference analysis was performed using MrBayes [24] which relies on a modified version of the Juke–Cantor model for morphological data as outlined by Lewis [25], with variable character states

**Table 1.** List of species added to the matrix of Thuy & Stöhr [6,9] used to explore the phylogenetic position of *Ophiogojira* gen. nov. and *Ophioduplantiera* gen. nov.

| taxon | author | family assignment | age |
|---|---|---|---|
| *Ophiura ooplax* | H.L. Clark (1911) | Ophiuridae | recent |
| *Stegophiura sladeni* | Duncan (1879) | Ophiopyrgidae | recent |
| *Aspidophiura forbesi* | Duncan (1879) | Ophiopyrgidae | recent |
| *Amphiophiura radiata* | Lyman (1878) | Ophiopyrgidae | recent |
| *Ophiomusa simplex* | Lyman (1878) | Ophiomusaidae | recent |
| *Ophiomusa fallax* | Koehler (1904) | Ophiomusaidae | recent |
| *Ophiosphalma cancellatum* | Lyman (1878) | Ophiosphalmidae | recent |
| *Ophiosphalma glabrum* | Lütken & Mortensen (1899) | Ophiosphalmidae | recent |
| *Ophiolipus granulatus* | Koehler (1897) | Ophiosphalmidae | recent |
| *Ophioduplantiera noctiluca* | this study | Ophiuridae | Early Jurassic (late Sinemurian to late Pliensbachian) |
| *Ophiogojira labadiei* | this study | unknown | Early Jurassic (early Pliensbachian) |
| *Ophioculina hoybergia* | Rousseau & Thuy (2018) | Ophiopyrgidae (Rousseau *et al.* 2018) | Late Jurassic (middle Tithonian) |
| *Aspidophiura seren* | Ewin & Thuy (2017) | Ophiopyrgidae | Middle Jurassic (middle Callovian) |
| *Enakomusium weymouthiense* | Damon (1880) | unknown | Late Jurassic (early Oxfordian) |
| *Enakomusium whymanae* | Ewin & Thuy (2017) | unknown | Middle Jurassic (middle Callovian) |

from 2 to 10 [26,27]. Only variable characters were sampled, and we compensated for character selection bias by letting MrBayes search for parsimony informative characters (Mkpars model) [27]. All character states were assumed to have equal frequency, and prior probabilities were equal for all trees.

We assumed that evolutionary rates vary between sites according to a discrete gamma distribution. In contrast with Thuy & Stöhr [6,9], however, we used a compound dirichlet prior rather than an exponential distribution on branch lengths, in accordance with previous recommendations [28,29]. Average standard deviations of split frequencies stabilized at about 0.007 after 3 million generations (mgen), sampled every 1000 generations. The first 25% of the trees were discarded as burnin. The consensus trees were examined with the software FigTree v. 1.4.4 by Rambaut (http://tree.bio.ed.ac.uk/software/figtree/). As is a common standard in statistics, we regard confidence intervals of 95–99% as strong support for a node to be true, and at least 90% probability as good support.

In addition to the Bayesian analysis, we ran a maximum likelihood (ML) and a parsimony analysis. The morphological phylogeny from 64 taxa and 134 characters was performed using an ML inference. The ML analysis was performed in IQ-TREE's v. 1.6.9 [30] with an ultrafast bootstrap (UFboo, -st Morph, -m MK + ASC). The MK model was selected (similar to a Jukes–Cantor model) and the ascertainment bias correction model (ASC) was applied to correct the branch lengths for the absence of constant sites. The resulting consensus tree was visualized using the software FigTree v. 1.4.4. The tree was rooted to *Aganaster gregarius*. Out of the 134 characters used for the construction of the morphological phylogeny, only 133 were parsimony informative and analysed in this study.

For the parsimony analysis, the matrix was analysed using PAUP* (v. 4.0a) [31] by performing a heuristic search (10 000 replicas) using the tree bisection–reconnection (TBR) algorithm; statistical support was obtained by performing 1000 bootstrap replicates and searching for successively longer trees to calculate decay indices. All characters were treated as unordered and of equal weights. The heuristic search resulted in 252 most parsimonious trees (tree length: 972 steps; consistency index = 0.218; retention index = 0.622).

## 2.3. Nomenclatural acts

The electronic edition of this article conforms to the requirements of the amended International Code of Zoological Nomenclature, and hence the new names contained herein are available under that Code from the electronic edition of this article. This published work and the nomenclatural acts it contains have been registered in ZooBank, the online registration system for the ICZN. The ZooBank LSIDs (Life Science Identifers) can be resolved and the associated information viewed through any standard web browser by appending the LSID to the prefix 'http://zoobank.org/'. The LSID for this publication is: urn:lsid:zoobank.org:pub:40958BC5-5766-43F4-A3C2-6EBD5239FD74; for *Ophiogojira*: urn:lsid:zoobank.org:act:71D3EFC7-C478-4657-858F-CA1E92F4E72B; for *Ophiogojira andreui*: urn:lsid: zoobank.org:act:8DA86259-8368-4FBC-B798-2A16F96D3103; for *Ophiogojira labadiei*: urn:lsid:zoobank. org:act:A0A50201-62E8-4A56-B88E-3486C30CEBDC; for *Ophioduplantiera*: urn:lsid:zoobank.org:act:5D A50C05-751A-4D63-9F2D-46F116A8F147; for *Ophioduplantiera noctiluca*: urn:lsid:zoobank.org:act: FFB9607F-E916-418D-99B5-7B38D40AE9AF. The electronic edition of this work was published in a journal with an ISSN, and has been archived and is available from the following digital repositories: PubMed, Central, LOCKSS.

# 3. Results

## 3.1. Phylogeny

The Bayesian inference analysis produced a well-resolved tree (figure 3) with relatively good support values. It shows the basal split of the living Ophiuroidea in the two superorders Ophintegrida and Euryophiurida, although support values of the latter are below 90%. Topology within the Ophintegrida largely agrees with previous results [6,9]. The tree fails to resolve the relationships at the base of the Euryophiurida, in particular the split between the orders Euryalida and Ophiurida. The sampled Mesozoic and living euryalids form a well-supported clade, in agreement with previous results [9]. Members of the ophiurid suborder Ophiomusina fall into a robust but unresolved clade with all three sampled members of the family Ophiomusidae, and a poorly supported clade with all sampled members of the family Ophiosphalmidae and the two fossil *Enakomusium* species.

The suborder Ophiurina forms a poorly supported clade with *Ophiogojira labadiei* gen. et sp. nov. at its base. Members of the family *Ophiopyrgidae* are split between a robust *Aspidophiura* clade and a moderately well-supported clade formed by *Stegophiura sladeni* and *Palaeocoma milleri* in a poorly supported sister relationship with *Amphiophiura radiata*. Members of the family Ophiuridae form a well-supported clade including the two fossils *Ophioduplantiera noctiluca* gen. et sp. nov. and *Ophioculina hoybergia*.

The ML analysis produced a tree (figure 4) with moderate to good support values. Its topology agrees with that of the Bayesian inference tree when it comes to the superorder-level splits and the relationships within the Ophintegrida. Within the Euryophiurida, the euryalids form a robust clade nested within a poorly supported ophiomusin clade. The Ophiurina clade gains high overall support with *Ophiogojira labadiei* gen. et sp. nov in a robust sister relation to all other members of the suborder. Topology within the suborder is congruent with the Bayesian inference but with higher support values.

The parsimony analysis produced a moderately well-resolved tree (figure 5) that agrees mostly with the results obtained by Bayesian and ML analyses, with a few exceptions, most notably the position of the euryalid clade. The parsimony-based tree confirms the position of *Ophiogojira labadiei*, sister to the Ophiurina, and of *Ophioduplantiera noctiluca* within the Ophiuridae, albeit with poor support.

## 3.2. Systematic palaeontology

Class Ophiuroidea Gray, 1840
Subclass Myophiuroidea Matsumoto, 1915
Infraclass Metophiurida Matsumoto, 1913 (crown-group of Ophiuroidea)
Superorder Euryophiurida O'Hara *et al.* (2017)
Order Ophiurida Müller & Troschel (1840)
Suborder Ophiurina Müller & Troschel, 1840 *sensu* O'Hara *et al.* [5]
Family unknown
Genus *Ophiogojira* gen. nov.
Type species: *Ophiogojira labadiei* sp. nov., by present designation.

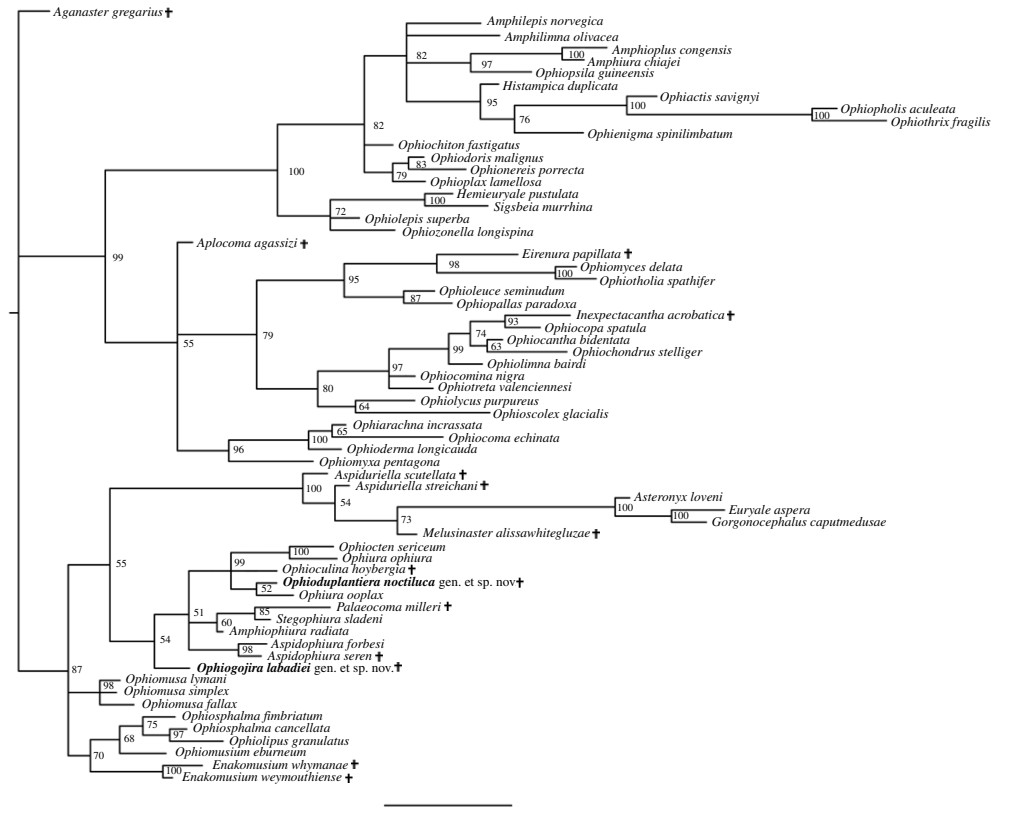

0.3

**Figure 3.** Bayesian phylogenetic tree of the matrix of Thuy & Stöhr [9] including the extinct species described herein (*Ophiogojira labadiei* gen. et sp. nov. and *Ophioduplantiera noctiluca* gen. et sp. nov.) marked in bold. Numbers at nodes indicate posterior probabilities. Extinct species are marked by a cross.

Diagnosis: Ophiurin ophiuroid with disc covered by small scales and large radial shields; row of contiguous, block-shaped papillae along the genital slit, extending until the distal tip of the abradial genital plate at the dorsoventral midline of the arms; second oral tentacle pore opening deep inside the mouth slit; arms cylindrical, with pentagonal dorsal arm plates in contact in proximal arm segments; ventral arm plates diamond-shaped, separated by lateral arm plates; rounded rectangular lateral arm plates with finely tuberculated outer surface and proximal edge lined by numerous well-defined spurs; small spine articulations integrated into outer surface stereom, with a lip-shaped vertical ridge separating the muscle and nerve openings; short, slightly flattened arm spines; tentacle pores developed as between-plate openings in proximal segments, and within-plate perforations in median to distal segments.

Etymology: Genus named in honour of French metal band Gojira, for producing songs of an unfathomable intensity, beautifully dark and heavy, and exploring the abyss of life and death, of human strength and error, and of thriving and yet threatened oceans.

Gender: feminine

Other species included: *Ophiogojira andreui* sp. nov.

*Ophiogojira labadiei* sp. nov.

Figures 6 and 7

Etymology: Species named after Jean-Michel Labadie, bass player of French metal band Gojira.

Holotype: MnhnL OPH159

Type locality and stratum: succession of clays and thin lumachellic beds with siderite nodules, dated to the late Early Pliensbachian (Davoei chronozone, Early Jurassic), near Sedan, Ardennes, France.

Paratypes: MnhnL OPH160–OPH166

Diagnosis: Species of *Ophiogojira* with large, high lateral arm plates, showing up to nine well-defined spurs along outer proximal edge with the two to three ventralmost spurs separated from the others by a larger gap; outer surface stereom with very fine tuberculation; two small spine articulations in proximal lateral arm plates (three in median lateral arm plates) tightly grouped close to ventro-distal tip of the plate; up to seven small spine articulations in distal lateral arm plates, with the three dorsalmost much smaller than the remaining spine articulations.

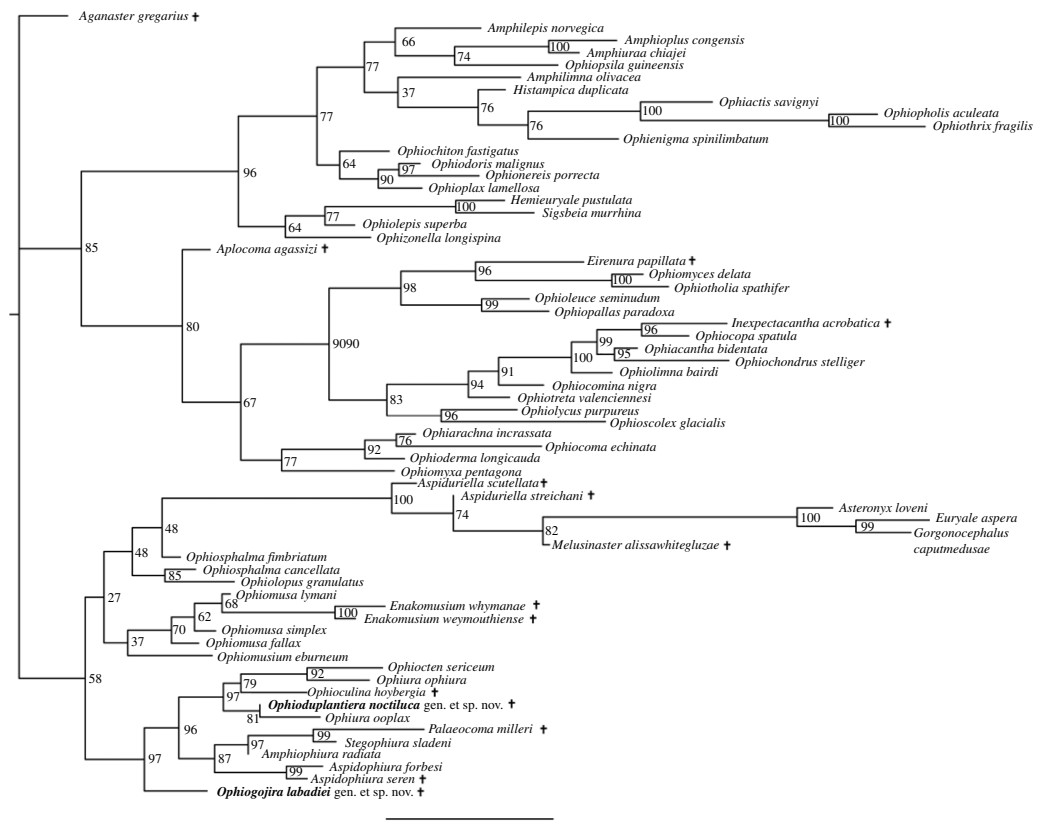

**Figure 4.** ML phylogenetic tree of the matrix of Thuy & Stöhr [9] including the extinct species described herein (*Ophiogojira labadiei* gen. et sp. nov. and *Ophioduplantiera noctiluca* gen. et sp. nov.) marked in bold. Numbers at nodes indicate bootstraps values. Extinct species are marked by a cross.

Description of holotype: MnhnL OPH159 is a fragment of an articulated disc (figure 6) preserving almost an entire radius, including a series of proximal arm segments, and exposing both the ventral and dorsal sides; disc fragment found in close association with a similarly preserved and similarly sized articulated proximal to the median arm fragment probably belonging to the same individual; dorsal side of the disc covered by large, rounded isosceles-triangular radial shields (figure 6*b*), entirely separated radially and interradially by multiple rows of very small to medium-sized, radially elongated, drop- to lens-shaped disc scales, devoid of granules or spines; radial shields exposing almost their entire external surface and accounting for approximately half of the disc radius; central primary plates not distinguishable; abradial genital plate with enlarged distal edge exposed below the distal edge of the radial shield, extending to dorsoventral midline of the arms (figure 6*c*), showing a row of tightly contiguous, block-shaped papillae; size of papillae on abradial genital plate increasing towards distalmost tip of genital plate, decreasing dorsalwards and ventro-proximalwards; row of papillae continuously and uniformly extending along the entire length of genital slit and around the corner formed by the oral and adoral shields (figure 6*a*); ventral interradius covered by rounded polygonal plates of variable size, devoid of granules or spines; oral shield broken, originally rounded pentagonal with shallow lateral incision forming an angle with the adoral shield and bearing small papillae; adoral shield L-shaped, larger distally, with small papillae along the distal-abradial edge; a single large, pointed, proximalwards bent oral papilla s.l. possibly corresponding to an adoral shield spine on the distal-adradial edge, tightly interlocking with a first ventral arm plate with a pointed proximal tip, concave lateral edges and a convex distal edge; no other oral papillae preserved.

Disc fragment preserving arm stump comprising seven segments; arm oval in cross-section; dorsal arm plates pentagonal, as wide as long, overlapping and devoid of a conspicuous ornamentation; ventral arm plates roughly pentagonal, with a broad, rectangular distal portion, small, latero-proximal incisions for the tentacle pore, and a pointed proximal tip with concave edges, separated by lateral arm plates; tentacle openings small, encompassed by ventral and lateral arm plates in all seven segments (figure 6*a*), bordered by a weak ridge on the lateral arm plate, covered by two roughly

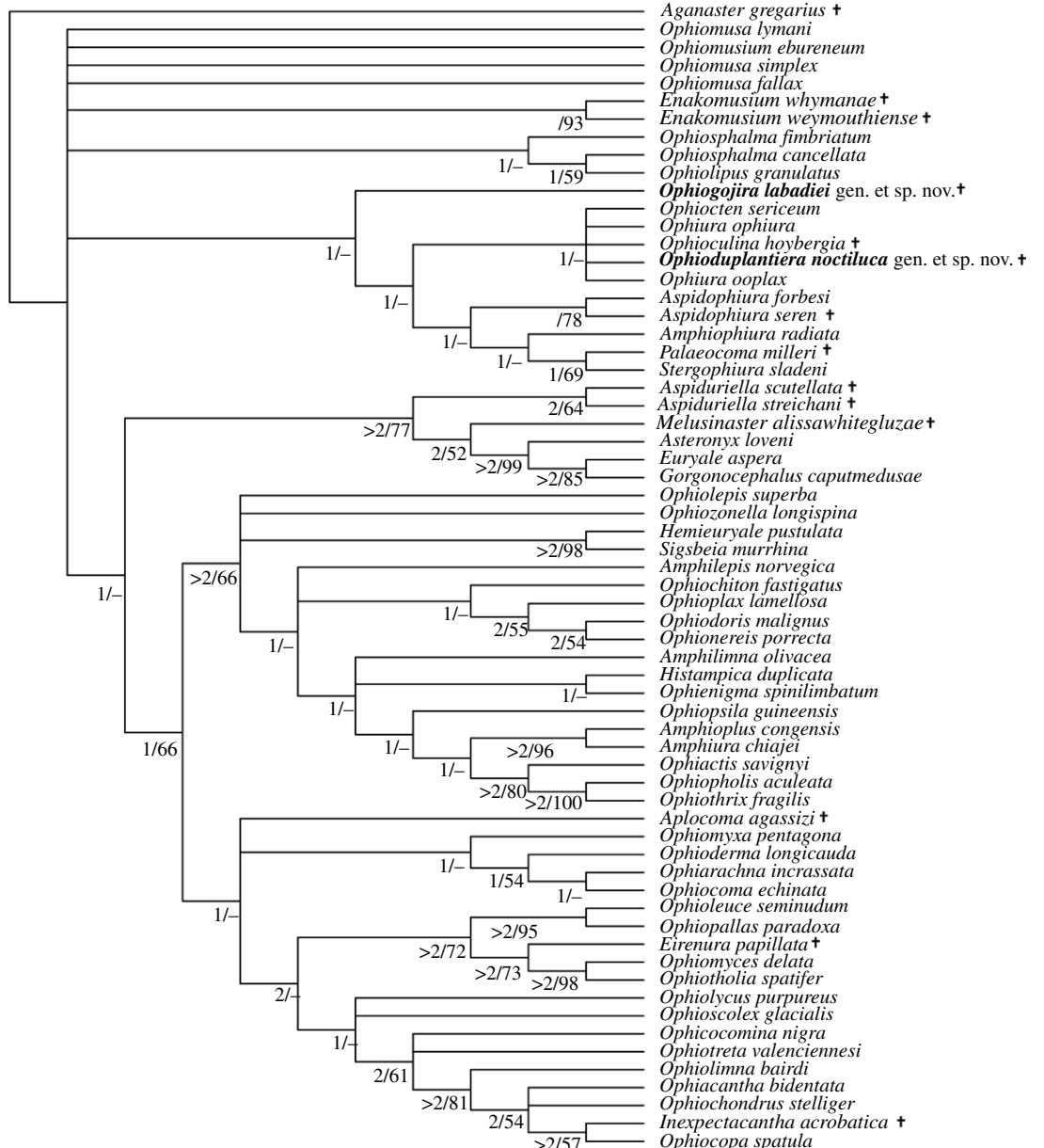

**Figure 5.** Consensus tree derived from the parsimony analysis in PAUP of the matrix of Thuy & Stöhr [9] including the extinct species described herein (*Ophiogojira labadiei* gen. et sp. nov. and *Ophioduplantiera noctiluca* gen. et sp. nov.) marked in bold. Numbers at nodes indicate decay indices and bootstrap values, respectively. Extinct species are marked by a cross.

semicircular tentacle scales; lateral arm plates stout, higher than long, with a straight distal edge and pointed dorsal and ventral tips; outer surface stereom with trabecular intersections transformed into very small tubercles; fine horizontal striation close to the proximal edge mostly hidden by overlapping distal edge of preceding lateral arm plate (figure 6*d*); two small spine articulations in the ventro-distal corner of the outer surface, integrated into the outer surface stereom, slightly sunken, separated from the distal edge by a wide area composed of more coarsely meshed stereom; spine articulations (figure 6*e*) pointing latero-distalwards, composed of a small muscle opening proximally bordered by a narrow, denticulate vertical ridge and distally bordered by a large, swollen, lip-shaped vertical ridge: slightly smaller nerve opening on ventro-distal edge of the large, lip-shaped vertical ridge; arm spines (figure 6*f*) small, approximately as long as one-third of an arm segment, adpressed against lateral arm plates, slightly flattened, lanceolate, with a blunt tip and a very fine, irregular longitudinal striation.

Associated arm fragment (figure 7*a*) preserving seven proximal to median segments; all plates as in arm stump of disc fragment, except for a gradually increasing length–width ratio; tentacle openings abruptly transformed from between-plate openings to within-plate perforations, opening below the

ventral spine articulation and covered by a single small, oval tentacle scale; two arm spine articulations in all preserved segments.

Paratype supplements: MnhnL OPH160 is an articulated disc (figure 7*b*–*e*) of a subadult individual exposing both dorsal and ventral sides and preserving proximalmost segments of all five arms; disc diameter 4.8 mm; disc round to very weakly pentagonal, flat; dorsal side of disc covered by numerous relatively thin, round scales of variable size; central primary plate (figure 7*b*) distinguishable but of similar size to surrounding disc scales, radial primary plates not discernible; radial shields as in holotype, accounting for almost half of the disc radius; ventral interradius as in holotype; oral shields (figure 7*c*,*e*) arrow-head-shaped, with clearly incised lateral edges and a slightly acute proximal angle; proximalwards bent potential adoral shield spine as in holotype, with proximally tightly interlocking additional oral papilla s.l. attached to adoral shields; four rectangular oral papillae s.l. (figure 7*e*) forming a tight row along the edge of the oral plate, in proximal continuation of two papillae attached to adoral shield; ventralmost tooth single, small, rounded conical. Arms as in holotype but with slightly longer arm segments and slightly coarser outer surface stereom; ridge bordering tentacle openings on lateral arm plates slightly better developed than in holotype.

MnhnL OPH163 is a dissociated proximalmost lateral arm plate (figure 7*h*–*i*); approximately 1.5 times higher than long, with rounded dorsal and ventral edges; distal edge weakly convex with incision close to spine articulations; proximal edge concave, lined by conspicuous band of more coarsely meshed, finely horizontally striated stereom, with six well-defined, oval, prominent and slightly protruding spurs (marked as SP in figure 7*h*) along the proximal edge, ventrally followed by a larger, oblique prominent and protruding spur, and two much smaller, weakly defined, non-prominent and non-protruding spurs; the six dorsal spurs separated from the three ventral ones by a larger gap; outer surface stereom with trabecular intersections transformed into very small tubercles; two small spine articulations close to the ventro-distal tip of the lateral arm plate (marked as SA in figure 7*h*), integrated into the outer surface stereom, slightly sunken and separated from distal edge by a wide area composed of more coarsely meshed stereom; spine articulations as in holotype; tentacle opening developed as deep, ventralwards-pointing incision lined by a sharply defined groove. The inner side of the lateral arm plate (figure 7*i*) with a single, large, well-defined, strongly prominent ridge composed of more densely meshed stereom and with a tongue-shaped dorsal tip; eight moderately well defined, weakly prominent horizontal, oval spurs along the inner side of the distal edge; no perforations discernible.

MnhnL OPH164 (figure 7*j*–*k*) and MnhnL OPH165 (figure 7*l*,*m*) are dissociated proximal to median lateral arm plates, slightly higher than long and almost as long as high, respectively, with straight dorsal edge and weakly convex ventral edge; proximal edge with band of more coarsely meshed, weakly horizontally striated stereom and spurs as in OPH163 except for slightly larger gap between dorsal and ventral groups of spurs; outer surface as in holotype and in OPH163; three spine articulations as in holotype and in OPH163, grouped together close to the ventro-distal tip of the lateral arm plate; tentacle pore developed as small within-plate opening ventrally bordering the ventralmost spine articulation. The ridge on the inner side (figure 7*k*,*m*) slightly wider than in OPH163 and with a wider dorsal tip; six spurs along inner side of distal edge; tentacle pore opening close to the ventro-distal tip of the ridge.

MnhnL OPH166 is a dissociated distal lateral arm plate (figure 7*n*–*o*), slightly longer than high; general similar to OPH164 and OPH165 but with four regular spurs on outer proximal edge in addition to larger, oblique spur close to ventro-proximal tip of lateral arm plate separated from the four regular spurs by a large, conspicuous gap; seven spine articulations, equal-sized except for the dorsalmost three; tentacle pore as in OPH164 and OPH165. The inner side with the ridge and tentacle pore as in OPH164 and OPH165, and with five spurs on the inner distal edge.

MnhnL OPH162 is a dissociated proximal vertebra (figure 7*g*), distal outline ovoid; dorso-distal muscle fossae dorsalwards converging and slightly distawards projecting; muscle fossae with pronounced growth lines; zygocondyles on distal surface dorsalwards converging; zygosphene projecting beyond ventral edge of zygocondyles, with projecting part approximately as long as zygocondyles.

MnhnL OPH161 is a dissociated oral plate (figure 7*f*), more than two times longer than high, with abradial muscle fossa developed as central depression, and with adradial muscle attachment area ventrally lining the abradial articulation area.

Remarks: The specimens described above show a combination of characters that are typical for the order Ophiurida as diagnosed by O'Hara *et al.* [23]: unbranched arms with finely ornamented lateral arm plates in the lateral position, bearing spine articulations with a muscle opening proximally

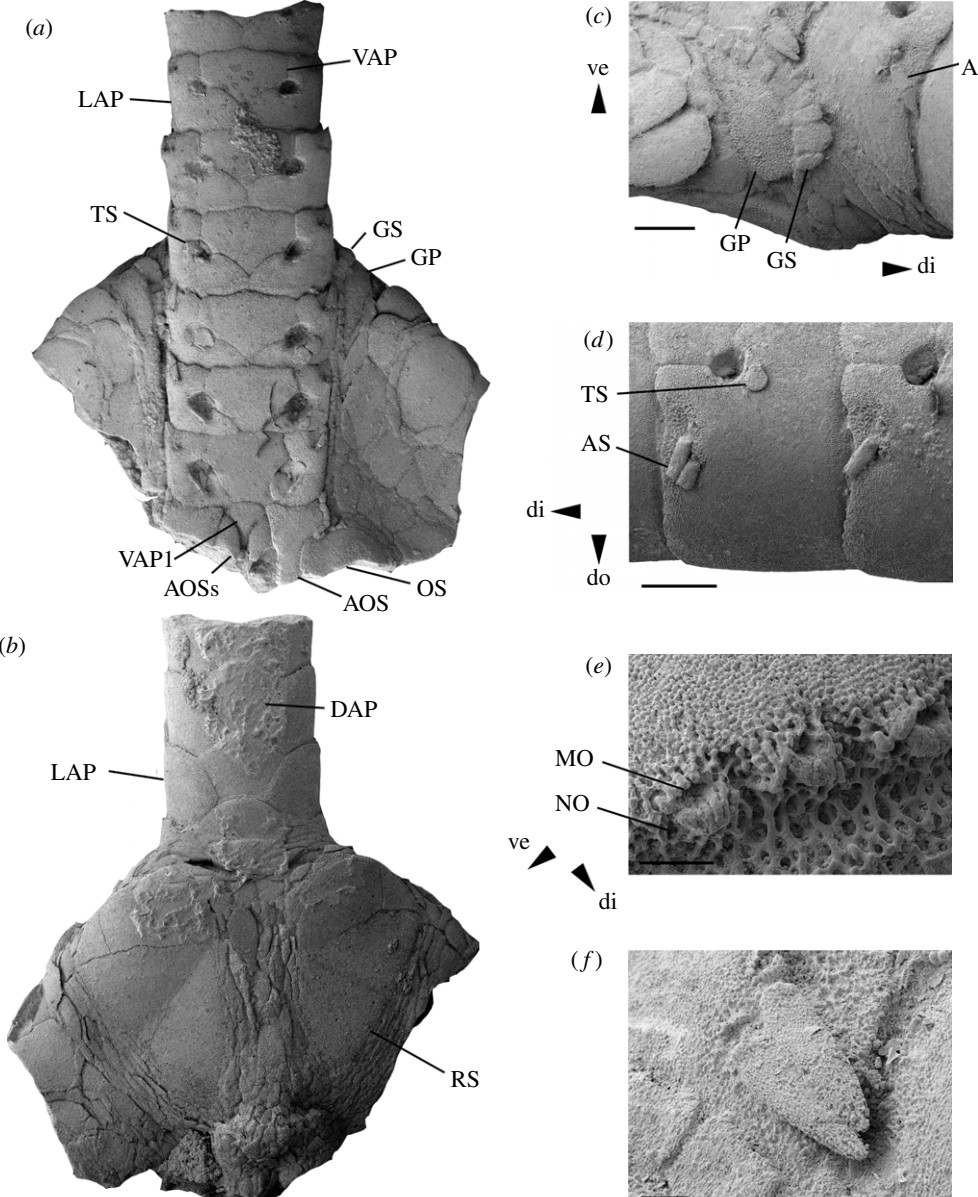

**Figure 6.** *Ophiogojira labadiei* gen. et sp. nov. from the late Early Pliensbachian (Davoei chronozone, Early Jurassic) of Sedan, Ardennes, France. (*a*–*f*) Holotype (MnhnL OPH159), disc fragment in ventral (*a*) and dorsal (*b*) views, and with details of the genital papillae (*c*), the proximal arm segments (*d*), the arm spine articulations (*e*) and the arm spines (*f*). AOS, adoral shield; AOSs, adoral shield spine; AS, arm spine; DAP, dorsal arm plate; di, distal; do, dorsal; GP, genital plate (abradial); GS, genital scale; LAP, lateral arm plate; MO, muscle opening; NO, nerve opening; RS, radial shield; TS, tentacle scale; VAP, ventral arm plate; VAP1, first ventral arm plate; ve, ventral. Scale bars equal 1 mm in (*a,b*), 0.5 mm in (*c,d*) and 0.1 mm in (*e,f*).

encompassed by a ridge and distally separated from the nerve opening by a vertical ridge. In spite of superficial similarities with members of the suborder Ophiomusina, in particular *Ophiomusa* Hertz, 1927, the specimens described above cannot be assigned to any living member of this suborder because the spine articulations are vertical rather than oblique, the tentacle pores are developed as between-plate openings until median arm segments, and the genital plates bear a continuous row of block-shaped papillae that extend until the distal tip of the abradial genital plate at the dorsoventral midline of the arms. The other suborder Ophiurina comprises the families Astrophiuridae, Ophiuridae and Ophiopyrgidae [5], of which only the latter two come into consideration because of the extreme skeletal modifications of the Astrophiuridae. The Ophiuridae and Ophiopyrgidae are difficult to distinguish unambiguously because most of the diagnostic characters show considerable intraspecific variation and/or exceptions [23]. The combination of characters of the specimens described above

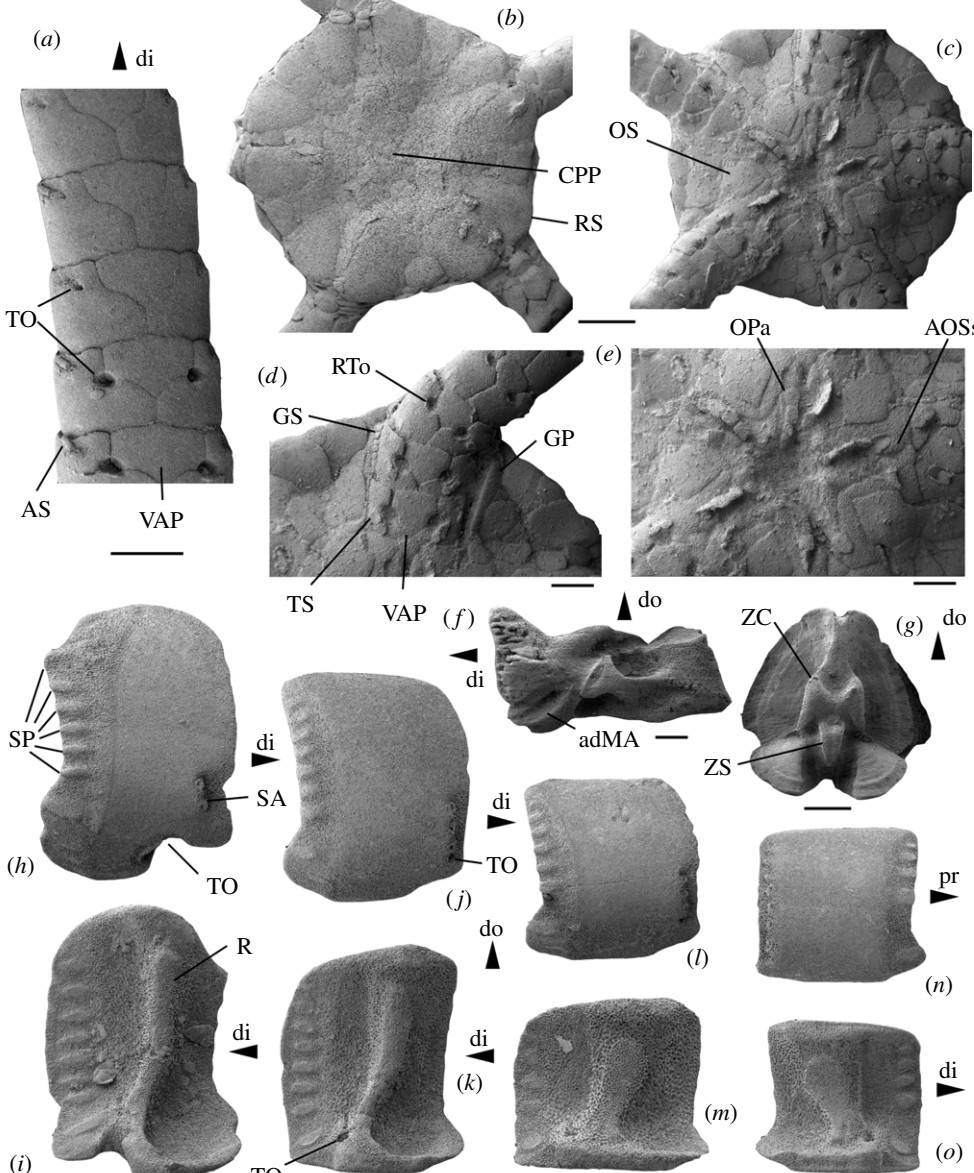

**Figure 7.** *Ophiogojira labadiei* gen. et sp. nov. from the late Early Pliensbachian (Davoei chronozone, Early Jurassic) of Sedan, Ardennes, France. (*a*) Holotype (MnhnL OPH159), arm fragment in ventral view. (*b–e*) Paratype (MnhnL OPH160), disc in dorsal (*b*) and ventral (*c*) views and with details of ventral arm base (*d*) and oral skeleton (*e*). (*f*) Paratype (MnhnL OPH161), oral plate in adradial view. (*g*) Paratype (MnhnL OPH162), proximal vertebra in distal view. (*h–i*) Paratype (MnhnL OPH163), proximalmost lateral arm plate in external (*h*) and internal (*i*) views. (*j–k*) Paratype (MnhnL OPH164), proximal lateral arm plate in external (*j*) and internal (*k*) views. (*l–m*) paratype (MnhnL OPH165), median lateral arm plate in external (*l*) and internal (*m*) views. (*n–o*) Paratype (MnhnL OPH166), distal lateral arm plate in external (*n*) and internal (*o*) views. adMA, adradial muscole attachment area; AOSs, adoral shield spine; AS, arm spine; CPP, central primary plate; di, distal; do, dorsal; GP, genital plate (abradial); GS, genital scale; OPa, oral papillae *sensu* lato; pr, proximal; R, vertebral articular ridge; RS, radial shield; RTo, ridge bordering the tentacle opening; SA, spine articulation; SP, spur; TO, tentacle opening; TS, tentacle scale; VAP, ventral arm plate; ZC, zygocondyle; ZS, zygosphene. Scale bars equal 1 mm in (*a–c*), and 0.5 mm in (*d–o*).

does not match any known extant genus, nor does it allow for a conclusive family-level assignment. This observation is corroborated by the results of the phylogenetic analysis (see above), suggesting the position of the specimens in question at the base of the Ophiuridae and Ophiopyrgidae. Therefore, we consider them as an extinct ophiurin member of an unresolved family-level placement.

The known fossil record of the Ophiurida includes only a few species that are seemingly similar to the specimens from Sedan described above. The orientation of the spine articulations and the development of the genital scales precludes assignment to any species of *Enakomusium* Thuy, 2015, to *Ophiomusium*

*murravii* (Forbes, 1843) from the Pliensbachian of Great Britain, and *Mesophiomusium paragraysonensis* (Taylor, 1966) from the Aptian of Antarctica, despite the similar development of the tentacle openings in these taxa. *Ophiomusium? solodurense* Hess, 1962 from the Pliensbachian of Switzerland has larger, slightly raised, oblique spine articulations arranged along the entire outer distal edge, tentacle pores developed as within-plate perforations even in the proximal lateral arm plates, and much larger contact surfaces with the opposite lateral arm plates on the internal side. *Aspidophiura? seren* Ewin & Thuy, 2017 from the Callovian of Great Britain has much larger spine articulations placed directly at the outer distal edge, and lacks spurs along the outer proximal edge. *Ophioculina hoybergia* Rousseau & Thuy, 2018 from the Tithonian of Spitsbergen has a fully developed arm comb, i.e. genital papillae extending not only to the dorsoventral midline of the arms but to the dorsal disc surface, longer arm spines and superficial second oral tentacle pores.

From the comparisons made above, we conclude that the Sedan specimens belong to a previously unknown genus and species of unresolved family-level position within the suborder Ophiurina. We introduce the new genus and species *Ophiogojira labadiei* to accommodate the specimens in question. Differences with the slightly younger congener *Ophiogojita andreui* gen. et sp. nov. from Sanem are discussed below. Because the Sedan material is outstandingly well preserved and comprises articulated disc fragments and arm portions, *Ophiogojira labadiei* is designated as the type species of the new genus.

*Ophiogojira andreui* sp. nov.

Figure 8*a–g*

Etymology: Species named after Christian Andreu, guitar player of French metal band Gojira.

Holotype: MnhnL OPH167

Type locality and stratum: dark grey marls, Ottempt Member of the Aubange Formation, earliest Toarcian (Tenuicostatum chronozone, Early Jurassic), Uerschterhaff near Sanem, Luxembourg.

Paratypes: MnhnL OPH168 and OPH169

Diagnosis: Species of *Ophiogojira* with moderately large and relatively low lateral arm plates; up to six spurs in a continuous row along the outer proximal edge without gaps; outer surface stereom with a fine tuberculation; three (proximal lateral arm plates) to four (median to distal lateral arm plates) small spine articulations along the outer distal edge.

Description of holotype: MnhnL OPH167 (figure 8*a,b*) is a dissociated proximal lateral arm plate, nearly as high as wide; dorsal and distal edges weakly convex to nearly straight; ventral edge almost straight except for shallow tentacle notch; proximal edge weakly concave with six well defined, prominent, protruding, oval and approximately equal-sized and evenly spaced spurs (marked as SP in figure 8*a*), with a slight increase in length from the dorsalmost to the ventralmost spur, and with a moderately well defined and slightly lowered area of finely horizontally meshed stereom distally bordering spurs; outer surface covered by very finely meshed stereom with trabecular intersections transformed into slightly enlarged, tightly spaced tubercles. Three very small spine articulations in a vertical row along the two ventral thirds of the distal edge, equal-sized, separated from the distal edge of the plate by a slightly thinned area lowered individually at every spine articulation and composed of more coarsely meshed stereom, approximately two times wider than the spine articulations; spine articulations composed of nearly equal-sized muscle and nerve openings separated by a large, well-defined, strongly protruding, lip-shaped vertical ridge; proximal edge of muscle opening tightly encompassed by thin but sharply defined and protruding, weakly denticulate ridge. Tentacle notch shallow but conspicuously indenting ventral edge of lateral arm plate.

The inner side of the lateral arm plate (figure 8*b*) with large, conspicuous, sharply defined, tongue-shaped central vertebral articulation composed of more finely meshed stereom; inner distal edge with six oval, well-defined, weakly prominent, equal-sized and equidistant spurs composed of more finely meshed stereom; tentacle opening developed as narrow, sharply defined, ventralwards-pointing notch; no other perforations discernible.

Paratype supplements: MnhnL OPH168 (figure 8*c–e*) is a dissociated median lateral arm plate slightly longer than high; nearly straight dorsal and ventral edges, weakly concave proximal edge and very slightly convex, almost straight distal edge, resulting in a rectangular general outline of the plate; ventral part of the lateral arm plate not protruding; six spurs as in holotype; outer surface ornamentation as in holotype. Four spine articulations (figure 8*e*) as in holotype but with distance between two middle spine articulations shortest; stereom between spine articulations and distal edge of the lateral arm plate slightly uniformly lowered rather than individually at every spine articulation as in holotype. Tentacle opening developed as small but conspicuous within-plate perforation on the ventral edge of the ventralmost spine articulation.

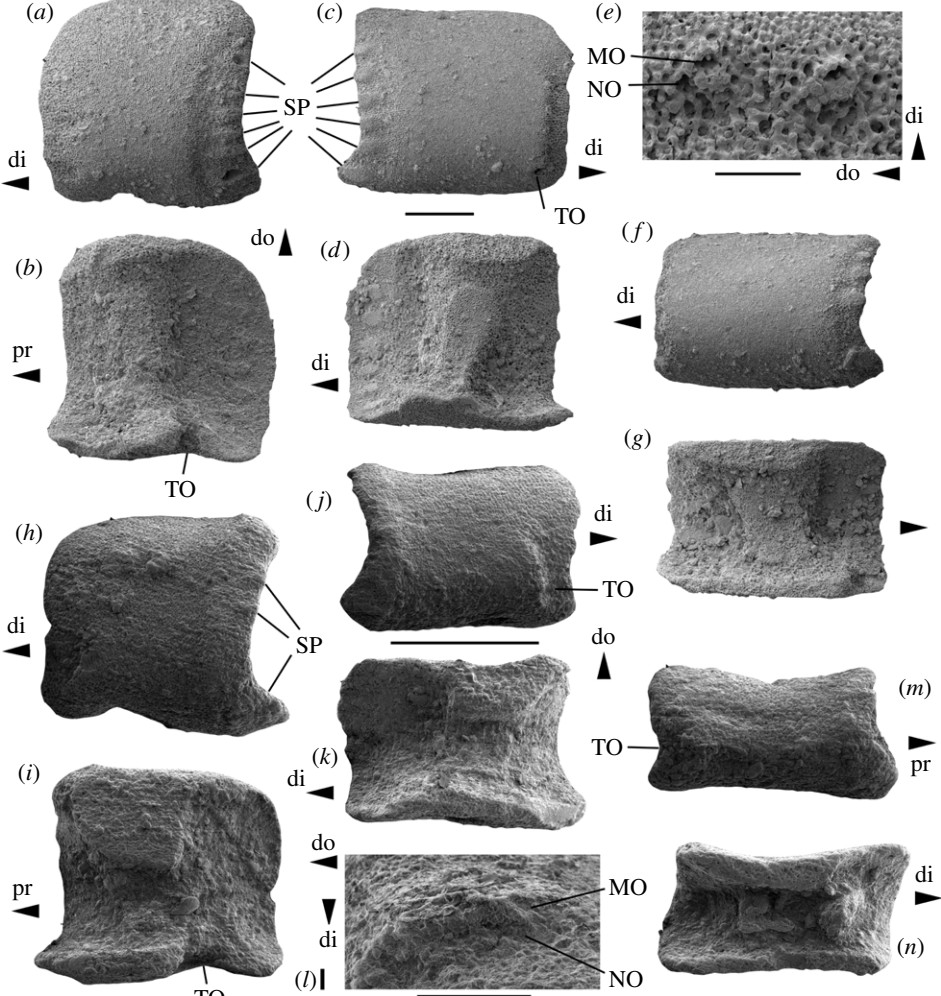

**Figure 8.** (*a*–*g*) *Ophiogojira andreui* gen. et sp. nov., from the earliest Toarcian (Tenuicostatum chronozone, Early Jurassic) of Uerschterhaff near Sanem, Luxembourg. (*a*,*b*) Holotype (MnhnL OPH167) proximal lateral arm plate in external (*a*) and internal (*b*) views. (*c*–*e*) Paratype (MnhnL OPH168), median lateral arm plate in external (*c*) and internal (*d*) views and with detail of spine articulations (*e*). (*f*–*g*) Paratype (MnhnL OPH169), distal lateral arm plate in external (*f*) and internal (*g*) views. (*h*–*n*) *Ophioduplantiera noctiluca* gen. et sp. nov., from the late Sinemurian to late Pliensbachian (Raricostatum to Margaritatus chronozones, Early Jurassic), Glasenbach Gorge near Salzburg, Austria. (*h*,*i*,*l*) Holotype (MnhnL OPH170) proximal lateral arm plate in external (*h*) and internal (*i*) views and with detail of spine articulations (*l*). (*j*,*k*) Paratype (MnhnL OPH171), median lateral arm plate in external (*j*) and internal (*k*) views. (*m*–*n*) paratype (MnhnL OPH172), distal lateral arm plate in external (*m*) and internal (*n*) views. di, distal; do, dorsal; MO, muscle opening; NO, nerve opening; pr, proximal; SP, spur; TO, tentacle opening. Scale bars equal 0.2 mm.

The inner side (figure 8*d*) with vertebral articulation as in holotype but slightly wider; spurs on inner distal edge as in holotype; tentacle opening obscured by matrix; ventral edge of the lateral arm plate most strongly protruding at the level of the vertebral articulation; no other perforation discernible.

MnhnL OPH169 (figure 8*f*–*g*) as a dissociated median to distal lateral arm plate, almost two times longer than high; well in agreement with holotype but with four spurs, the second ventralmost one probably broken (corresponding to the fifth spur) instead of six spurs on the outer proximal edge, with a shorter distance between the dorsalmost spine articulation and the dorsal edge of the lateral arm plate, and with a narrower, possibly incomplete band of stereom between the spine articulations and the distal edge of the lateral arm plate. The inner side of the lateral arm plate (figure 8*g*) with vertebral articulation as in holotype but more oblique and wider; five spurs on inner distal edge as in holotype; tentacle opening and other possible perforations on the inner side of the lateral arm plate obscured by matrix.

Remarks: The specimens described above are similar to those described from the Pliensbachian of Sedan and fully comply with the diagnosis of the new genus *Ophiogojira* introduced to accommodate

the material in question. The dissociated lateral arm plates from the Toarcian of Sanem differ from the slightly older ones from Sedan in the arrangement of the spurs along the outer proximal edge, and the number and arrangement of the spine articulations. These differences are sufficiently substantial to introduce a second species of *Ophiogojira*.

Family Ophiuridae Müller & Troschel, 1840

Genus *Ophioduplantiera* gen. nov.

Type and only known species: *Ophioduplantiera noctiluca* sp. nov., by present designation.

Diagnosis: Ophiurid genus with small lateral arm plates of rectangular outline, with up to four spurs on the outer proximal edge, and tiny tubercles on the outer surface arranged in a very weak vertical striation; three (proximal lateral arm plates) to two (median and distal lateral arm plates) with small spine articulations grouped near ventro-distal edge of the plate, and with a vertical ridge separating the muscle and nerve openings and a smooth ridge proximally bordering the muscle opening; inner side with large dorsal contact surface with opposite lateral arm plate; tentacle openings developed as ventralwards-pointing between-plate notch in proximal lateral arm plates, distalwards pointing within-plate perforation in the median lateral arm plates and distalwards pointing between-plate notch in distal lateral arm plates.

Etymology: Genus named after Joseph (Joe) and Mario Duplantier of French metal band Gojira, to honour their inspiring artistic achievements and their authentic, compassionate personalities.

Gender: feminine

*Ophioduplantiera noctiluca* sp. nov.

Figure 8*h–n*

Etymology: Species name formed from Latin 'noctiluca', literally translating into 'night light', in reference to the bioluminescent dinoflagellate genus *Noctiluca*, commonly called 'sea sparkle', because the name-bearing ophiuroid species brings light into the dark origins of the Ophiuridae.

Holotype: MnhnL OPH170

Type locality and stratum: marls from the Kehlbach and Scheck Members within the Adnet Formation, late Sinemurian to late Pliensbachian (Raricostatum to Margaritatus chronozones, Early Jurassic), Glasenbach Gorge near Salzburg, Austria.

Paratypes: MnhnL OPH171 and OPH172

Diagnosis: as in genus.

Description of holotype: MnhnL OPH170 (figure 8*h–i,l*) is a dissociated proximal lateral arm plate, approximately as long as high, with a very weakly concave dorsal edge, an incised ventral edge and a weakly convex distal edge; proximal edge uniformly concave, lined by a broad, slightly prominent band of more coarsely meshed stereom, with three small, horizontally elongate, poorly defined, slightly prominent and protruding spurs (marked as SP in figure 8*h*) in the dorsalmost third of the proximal edge, and with an additional larger, oblique, weakly prominent and protruding spur (marked as SP in figure 8*h*) close to the ventro-proximal tip of the lateral arm plate; outer surface stereom very finely meshed, with trabecular intersections transformed into tiny, closely spaced tubercles, showing a very weak tendency to form a faint vertical striation; two small spine articulations grouped close to ventro-distal tip of the lateral arm plate, separated from the distal edge by a sunken area of more coarsely meshed stereom approximately two times wider than the spine articulations; spine articulations (figure 8*l*) composed of small muscle and nerve openings separated by a well-defined, protruding, lip-shaped vertical ridge; proximal edge of muscle opening tightly encompassed by thin but sharply defined and protruding, smooth ridge. Ventral edge of the lateral arm plate with an incision corresponding to the tentacle notch.

The inner side of the lateral arm plate (figure 8*i*) with a conspicuous, large dorsal contact surface with the opposite lateral arm plate, and a much thinner ventral edge; small, inconspicuous, well-defined, oblique triangular vertebral articular ridge; two small, very weakly defined spurs close to the dorsal tip and a third one close to the ventral tip of the inner distal edge; tentacle opening developed as a well-defined, ventralwards-pointing notch.

Paratype supplements: MnhnL OPH171 (figure 8*j–k*) is dissociated median lateral arm plate, approximately 1.5 times longer than high, of rectangular outline, with a concave dorsal edge and a very weakly concave ventral edge; outer proximal edge as in holotype; outer surface stereom with a slightly better developed vertical striation; three small spine articulations as in holotype; ventralmost spine articulation distally bordered by a small tentacle perforation (marked as TO in figure 8*j*). Dorsal contact surface (figure 8*k*) slightly smaller than in holotype; vertebral articular ridge and spurs on the inner distal edge as in holotype; tentacle perforation small, inconspicuous, close to ventro-distal edge of the vertebral articular ridge.

MnhnL OPH172 (figure 8*m,n*) is a dissociated distal lateral arm plate, almost two times longer than high, of oblique rectangular outline; dorsal and ventral edges concave; proximal edge concave and lined by a poorly defined, slightly prominent band of more coarsely meshed stereom, with three small, horizontally elongate, poorly defined, slightly prominent and protruding spurs; distal edge deeply concave, incised by a distalwards pointing tentacle notch (marked as TO in figure 8*m*); outer surface stereom as in holotype; two very small spine articulations as in holotype, distally bordered by a lowered area of stereom at the same level as tentacle notch. Inner side with a relatively narrow dorsal contact surface; vertebral articular ridge small, triangular but better defined than in holotype; inner distal edge thinned at the level of the tentacle notch.

Remarks: The lateral arm plates described above unambiguously belong to the subclass Ophiurina because they are ornamented, arched and bearing spine articulations with muscle openings proximally bordered by a ridge and distally separated from the nerve opening by a slender, vertical ridge. The development of the tentacle opening is unusual: ventro-distalwards pointing between-plate tentacle openings in proximal lateral arm plates transform into distalwards pointing within-plate perforations in the median lateral arm plates that evolve into distalwards pointing between-plate openings in distal lateral arm plates. While the development and orientation of the tentacle openings in the proximal lateral arm plates are found in most ophiuroid groups [6], the distalwards pointing tentacle openings are typically found in the family Ophiuridae [23]. In line with the results of our phylogenetic analysis (see above), we, therefore, assign the material described above to the family Ophiuridae.

Because of the unusual development of the tentacle openings along the arm, the lateral arm plates described above differ from all fossil ophiuroid species known to date. This also includes the superficially similar species of *Enakomusium* and all other fossil members of the Ophiomusina, in which the tentacle openings never point distalwards, irrespective of whether they are developed as between-plate notches or within-plate perforations. The living members of the Ophiuridae all differ from the present material in having distalwards pointing tentacle openings also in proximal lateral arm plates. We, therefore, introduce the new genus and species *Ophioduplantiera noctiluca* to accommodate the specimens described above.

# 4. Discussion

The new fossil ophiuroids described in the present paper provide novel insights into the early evolution of the order Ophiurida. In the phylogenetic analyses, *Ophiogojira labadiei* gen. et sp. nov. holds a basalmost position within the suborder Ophiurina, sister to the Ophiuridae and Ophiopyrgidae. *Ophioduplantiera noctiluca* gen. et sp. nov., in contrast, has a more crownward position within the family Ophiuridae. Our analyses also included previously known fossil ophiuroids with assumed ophiurid affinities: *Palaeocoma milleri* and *Aspidophiura seren* both fall within the Ophiopyrgidae, and are in line with previous studies [6,32], whereas *Ophioculina hoybergia* belongs to the Ophiuridae, rather than the Ophiopyrgidae as suggested in the original description of the taxon [26]. The two species of *Enakomusium* are phylogenetically close to the Ophiosphalmidae and Ophiomusaidae, thus corroborating their inclusion in the suborder Ophiomusina, but our results are uncertain with respect to the relationships on family level.

Support values are reasonably high for morphology-based phylogenetic estimates and parallel those of previous analyses featuring incompletely known fossil species [6,9]. A comparison between Bayesian, ML and parsimony estimates confirms that parts of the tree are subject to a certain degree of variability and uncertainty, implying that topologies of some taxa should not be taken at face value. Nevertheless, the essential conclusions of the present study are the basal position of *Ophiogojira* at the stem of the Ophiuridae–Ophiopyrgidae clade and a more derived position of *Ophioduplantiera* within the Ophiuridae, which are resolved and well supported by both approaches.

The morphologies of *Ophiogojira* and *Ophioduplantiera* corroborate that the early evolution of major extant clades happens via extinct stem members with intermediate character states (figure 9). *Ophiogojira* shows a row of genital papillae that extends to the disc ambitus, i.e. the ventro-dorsal midline of the arm bases (figure 9*a*). This represents an intermediate state between the exclusively ventral row of genital papillae in some members of the Ophiomusina and the true arm comb of the Ophiurina formed by genital papillae extending on the dorsal side of the arm base (figure 9*b*).

The orientation of the tentacle notch is another case of remarkable morphological transitions. Whereas most Ophiopyrgidae have a ventro-distalwards pointing notch (figure 9*c,d*) like most other ophiuroids, the Ophiuridae typically have a distalward pointing tentacle opening [23] (figure 9*e,f*).

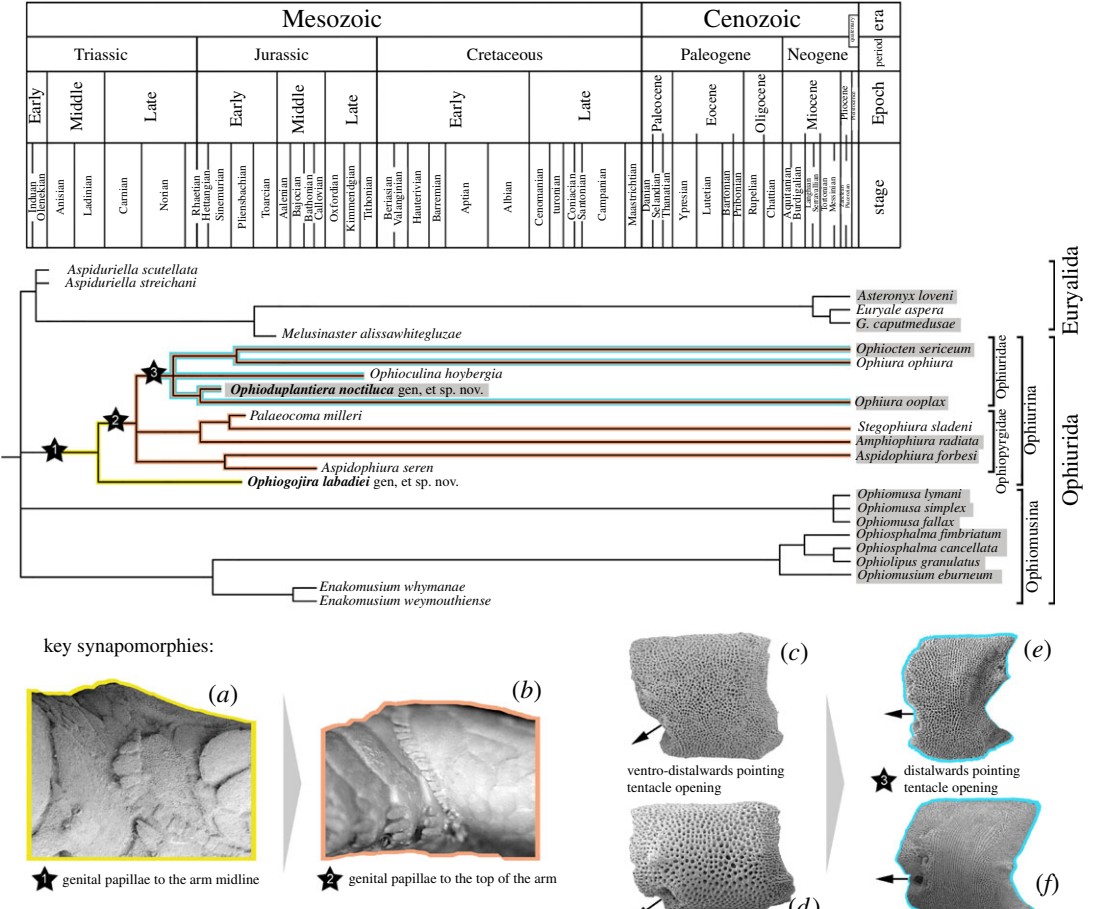

**Figure 9.** Evolutionary tree of the subclass Euryophiurida based on the Bayesian inference tree of figure 3, calibrated against the stratigraphic ranges of the fossil taxa, with key synapomorphies marked by stars and colour-mapped on the tree. Taxa shaded in grey predominantly occur below shelf depths (greater than 200 m). (*a*) Genital papillae in *Ophiogojira labadiei* gen. et sp. nov. in lateral view. (*b*) Genital papillae of *Ophiura ophiura* in lateral view. (*c–f*) Proximal lateral arm plates with dorsal side up and distal side to the left, of the ophiopyrgids *Amphiophiura radiata* (*c*) and *Aspidophiura forbesi* (*d*), and the ophiurids *Ophiura ophiura* (*e*) and *Ophiura ooplax* (*f*). Arrows show the orientation of tentacle openings.

Even in species with tentacle openings developed as within-plate perforations, like *Ophiura ooplax* (H.L. Clark, 1911), the opening points distalwards. The arm plates of *Ophioduplantiera* show a transition from ventralwards pointing between-plate notches in proximal plates, via distalwards pointing within-plate tentacle perforations in median plates, to distalwards pointing between-plate tentacle notches in the distal plates. *Ophioduplantiera* thus represents the oldest ophiuroid with distalward pointing tentacle openings and the oldest known member of the Ophiuridae in line with our phylogeny. The case of *Ophioduplantiera* furthermore suggests that the distalwards pointing tentacle opening evolved through a re-opening of a distalwards pointing within-plate tentacle perforation. In some recent ophiurid species, this transition seems to be reversed to some extent, with distalwards pointing tentacle notches transformed into distalwards pointing perforations in distal arm segments, like in *Ophiura trimeni* Bell, F. J. (1905), or in all arm segments, like in *Ophiura ooplax*.

Re-examination of *Ophioculina hoybergia* from upper Jurassic deep shelf sediments of Spitsbergen shows a similar transition from a ventralwards-pointing notch in the proximal arm segments to a distalwards pointing perforation in median ones. Whether this perforation opens into a distalwards pointing notch further along the arm in distal segments could not be determined unambiguously because of the unusual preservation of the *Ophioculina* material [33]. The oldest fossil ophiurid with a distalwards pointing between-plate tentacle notch in proximal arm segments was found in bathyal sediments of Aptian (early Cretaceous) age from Blake Nose in the Western tropical Atlantic [8].

Remarkably, the three oldest unambiguous members of the Ophiuridae are either from bathyal palaeo-depths (*Ophioduplantiera noctiluca* and the unnamed Aptian record from Blake Nose mentioned

above) or from boreal palaeo-latitudes isolated from the warm shelf seas at lower latitudes but assumedly connected with cold, deep waters (*Ophioculina hoybergia*). Because the coeval fossil record at shallower water depths and lower latitudes is disproportionately better sampled (e.g. [1]), we assume that the bias in the early fossil record of the Ophiuridae represents a true pattern and, therefore, conclude that the early evolution of the family took place in the ancient deep sea at bathyal palaeo-depths and in boreal deep shelf settings. This is in line with phylogenomic-based ancestral state reconstructions suggesting a deep to eurybathic early cladogenesis of the Ophiuridae [3]. Our results thus corroborate that the early evolution of some marine invertebrate clades can only be fully understood when deep-sea fossils are taken into account.

Data accessibility. All data used are included in the main article or have been uploaded as part of the electronic supplementary material.

The data are provided in electronic supplementary material [34].

Authors' contributions. B.T. collected, processed, picked and scanned the fossils, ran the Bayesian analysis, participated in the design of the study and drafted the paper; L.D.N.-T. participated in the fossil collecting and processing, assembled the figures and participated in the design of the study; T.P.-E. ran the Maximum-likelihood analysis and parsimony analyses and critically revised the manuscript. All authors gave final approval for publication and agree to be held accountable for the work performed therein.

Competing interests. We declare we have no competing interests.

Funding. No funding received for this contribution.

Acknowledgements. We thank Andy S. Gale (Portsmouth, UK) for help during fieldwork, and Sabine Stöhr (Stockholm, Sweden) for discussions on the morphology and phylogeny of *Ophiogojira*. We furthermore thank the reviewers whose comments greatly improved an earlier version of this manuscript.

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
