## [Peer Review File · Royal Society Open Science]

Review History

RSOS-210643.R0 (Original submission)

Review form: Reviewer 1 (Sabine Stöhr)

Is the manuscript scientifically sound in its present form?

Yes

Are the interpretations and conclusions justified by the results?

Yes

Is the language acceptable?

Yes

Do you have any ethical concerns with this paper?

No

Have you any concerns about statistical analyses in this paper?

No

Recommendation?

Accept with minor revision (please list in comments)

Comments to the Author(s)

This study is mainly a description of three new species and two new genera of brittle stars from fossil material. The descriptions are detailed and follow common practice in the field. The authors performed phylogenetic analyses to place the new genera in the current phylogenetic context. Both trees resolve the Ophiopyrgidae and Ophiuridae and place the new species in about the same relationship to these families. The relationship of these to the remaining groups in Euryophiurida is not resolved by either tree, but the Bayesian tree is closer to the latest molecular tree (Christodoulou et al. 2019), which must be considered the most likely hypothesis due to it being based on a much larger dataset. It is not surprising that Jurassic species do not fit with any of the recent families, but it is interesting how well they can be fitted in the tree. I agree that they are representatives of earlier lineages within these clades. Fossils contribute evolutionary information that DNA cannot resolve.

I found a few minor typographic errors:

p. 4 l 23: edged should be edges

p. 4 ls 44-45: delete Rambaut and URL, as you mentioned that already above

p. 11 l. 29/30: "belong" should be "belongs"

Table 1: should be ...matrix "of" Thuy and Stöhr., because the species were not added by Thuy and Stöhr

Fig. 2 caption: I think it should be "phylogeographic" instead of "phylogeographical"

In the Bayesian tree (fig. 3), a few numbers seem to overlap.

Review form: Reviewer 2 (Andy Gale)

Is the manuscript scientifically sound in its present form?

Yes

Are the interpretations and conclusions justified by the results?

Yes

Is the language acceptable?

Yes

Do you have any ethical concerns with this paper?

No

Have you any concerns about statistical analyses in this paper?

No

Recommendation?

Accept as is

Comments to the Author(s)

This paper continues a succession of important studies on the geological history of the deep-sea benthos, based upon Thuy's discovery that the fossil lateral arm plates of ophiuroids (an abundant living group) provide an accurate picture of the deep phylogeny of the group that is congruent with molecular studies. The paper is important because it demonstrates that the basal

Ophiurida had already colonised the deep sea by the Early Jurassic, and thus provides more evidence refuting the model that deep sea faunas underwent successive extinctions and recolonisation from shallower water habitats. It provides more fossil evidence of the deep history of the group.

I cannot find any faults with the paper, which is well written and cogently argued. I recommend publication

Decision letter (RSOS-210643.R0)

Dear Dr Thuy

On behalf of the Editors, we are pleased to inform you that your Manuscript RSOS-210643 "New fossils of Jurassic ophiurid brittle stars (Ophiuroidea; Ophiurida) provide evidence for early clade evolution in the deep sea" has been accepted for publication in Royal Society Open Science subject to minor revision in accordance with the referees' reports. Please find the referees' comments along with any feedback from the Editors below my signature.

Please submit your revised manuscript and required files (see below) no later than 7 days from today's (ie 21-Jul-2021) date. Note: the ScholarOne system will 'lock' if submission of the revision is attempted 7 or more days after the deadline. If you do not think you will be able to meet this deadline please contact the editorial office immediately.

on behalf of Dr Jeffrey Thompson (Associate Editor) and Kevin Padian (Subject Editor)
openscience@royalsociety.org

Associate Editor Comments to Author (Dr Jeffrey Thompson):

Comments to the Author:

Dear Ben et al.

Two expert reviewers, as well as myself acting as a third reviewer, have now seen your manuscript and provided comments. In general, all parties thought this was an excellent contribution worthy of publication in Open Science. That said, everything can be improved to some extent, and one of the reviewers has provided a few minor comments which should be dealt with to improve the manuscript. In addition, the reviewer noted in the comments to editor that although there is a nexus file supplied, really, a data matrix showing the characters and character states of the new taxa in relation to those of previous taxa, should also be included. In addition, after looking over the manuscript myself, I recommend adding some additional details concerning the analyses to the manuscript, as well as running a few additional analyses. All of the analytical details for the Bayesian methodology are listed in a previous paper (Thuy and Stohr, 2016). Instead of sending the readers back to that paper, I really think you should include the analytical details of your tree inference procedure in the manuscript. In addition to this, in looking through the nexus file, I noticed that the prior on branch lengths was an exponential distribution. These priors have been shown to be problematic ((Rannala et al. 2012, Zhang et al. 2012; see references below), and I would recommend running an additional Bayesian analysis using a compound dirichlet prior on branch lengths (`brlenspr=unconstrained:GammaDir(1.0,0.100,1.0,1.0)`), either in addition to or instead of, the exponential prior. In doing so with my own analyses, I do often get different results, so its worth considering running this additional analysis. I am also curious as to why you've chosen to perform a maximum likelihood analysis, instead of a parsimony analysis, as ML analyses of morphological data using the MK model have been shown to underperform parsimony analyses in terms of accuracy (Puttick et al. 2017, below). I would suggest you also run a parsimony analysis, in addition to your model-based approaches. All of that said, I applaud your use of Bayesian approaches, which have been shown time and again to be superior to parsimony, and I'm glad that the echinoderm community has embraced these novel, superior methods.

Additional notes:

Page 5, line 24. You note that your ML tree produced a fully resolved tree. Its worth noting that all ML trees are always fully resolved, so you should consider presenting this result differently in the manuscript.

Love the names! As always.

Jeff

Zhang, Chi, Bruce Rannala, and Ziheng Yang. "Robustness of compound Dirichlet priors for Bayesian inference of branch lengths." *Systematic Biology* 61.5 (2012): 779-784.

Rannala, Bruce, Tianqi Zhu, and Ziheng Yang. "Tail paradox, partial identifiability, and influential priors in Bayesian branch length inference." *Molecular biology and evolution* 29.1 (2012): 325-335.

Puttick, Mark N., et al. "Uncertain-tree: discriminating among competing approaches to the phylogenetic analysis of phenotype data." *Proceedings of the Royal Society B: Biological Sciences* 284.1846 (2017): 20162290.

Reviewer comments to Author:

Reviewer: 1

Comments to the Author(s)

This study is mainly a description of three new species and two new genera of brittle stars from fossil material. The descriptions are detailed and follow common practice in the field. The authors performed phylogenetic analyses to place the new genera in the current phylogenetic context. Both trees resolve the Ophiopyrgidae and Ophiuridae and place the new species in about the same relationship to these families. The relationship of these to the remaining groups in Euryophiurida is not resolved by either tree, but the Bayesian tree is closer to the latest molecular tree (Christodoulou et al. 2019), which must be considered the most likely hypothesis due to it being based on a much larger dataset. It is not surprising that Jurassic species do not fit with any of the recent families, but it is interesting how well they can be fitted in the tree. I agree that they are representatives of earlier lineages within these clades. Fossils contribute evolutionary information that DNA cannot resolve.

I found a few minor typographic errors:

p. 4 l 23: edged should be edges

p. 4 l s 44-45: delete Rambaut and URL, as you mentioned that already above

p. 11 l. 29/30: "belong" should be "belongs"

Table 1: should be ...matrix "of" Thuy and Stöhr., because the species were not added by Thuy and Stöhr

Fig. 2 caption: I think it should be "phylogeographic" instead of "phylogeographical"

In the Bayesian tree (fig. 3), a few numbers seem to overlap.

Reviewer: 2

Comments to the Author(s)

This paper continues a succession of important studies on the geological history of the deep-sea benthos, based upon Thuy's discovery that the fossil lateral arm plates of ophiuroids (an abundant living group) provide an accurate picture of the deep phylogeny of the group that is congruent with molecular studies. The paper is important because it demonstrates that the basal Ophiurida had already colonised the deep sea by the Early Jurassic, and thus provides more evidence refuting the model that deep sea faunas underwent successive extinctions and recolonisation from shallower water habitats. It provides more fossil evidence of the deep history of the group.

I cannot find any faults with the paper, which is well written and cogently argued. I recommend publication

===PREPARING YOUR MANUSCRIPT===

===PREPARING YOUR REVISION IN SCHOLARONE===

-- Ensure that your data access statement meets the requirements at <https://royalsociety.org/journals/authors/author-guidelines/#data>. You should ensure that you cite the dataset in your reference list. If you have deposited data etc in the Dryad repository, please only include the 'For publication' link at this stage. You should remove the 'For review' link.

Author's Response to Decision Letter for (RSOS-210643.R0)

See Appendix A.

Decision letter (RSOS-210643.R1)

Dear Dr Thuy,

I am pleased to inform you that your manuscript entitled "New fossils of Jurassic ophiurid brittle stars (Ophiuroidea; Ophiurida) provide evidence for early clade evolution in the deep sea" is now accepted for publication in Royal Society Open Science.

You can expect to receive a proof of your article in the near future. Please contact the editorial office (openscience@royalsociety.org) and the production office (openscience_proofs@royalsociety.org) to let us know if you are likely to be away from e-mail contact – if you are going to be away, please nominate a co-author (if available) to manage the proofing process, and ensure they are copied into your email to the journal. Due to rapid publication and an extremely tight schedule, if comments are not received, your paper may experience a delay in publication.

on behalf of Dr Jeffrey Thompson (Associate Editor) and Kevin Padian (Subject Editor)
openscience@royalsociety.org

Appendix A

Dear Jeff, dear referees,

Thank you for your time and for your constructive comments. All three of us were pleased to see that you were so positive about our submission. Please find below a point-by-point reply to the comments raised (quotes in *italics*). All three of us agree that the issues raised have improved our paper considerably. Therefore, it was our pleasure to implement the suggested changes.

Editor:

The reviewer noted in the comments to editor that although there is a nexus file supplied, really, a data matrix showing the characters and character states of the new taxa in relation to those of previous taxa, should also be included.

We added a character matrix to the Supplementary Material.

In addition, after looking over the manuscript myself, I recommend adding some additional details concerning the analyses to the manuscript, as well as running a few additional analyses. All of the analytical details for the Bayesian methodology are listed in a previous paper (Thuy and Stohr, 2016). Instead of sending the readers back to that paper, I really think you should include the analytical details of your tree inference procedure in the manuscript.

We agree and added all relevant analytical details.

In addition to this, in looking through the nexus file, I noticed that the prior on branch lengths was an exponential distribution. These priors have been shown to be problematic ((Rannala et al. 2012, Zhang et al. 2012; see references below), and I would recommend running an additional Bayesian analysis using a compound dirichlet prior on branch lengths ($\text{brlenspr}=\text{unconstrained:GammaDir}(1.0,0.100,1.0,1.0)$), either in addition to or instead of, the exponential prior. In doing so with my own analyses, I do often get different results, so its worth considering running this additional analysis.

Thank you for pointing out this issue, which we were not aware of. We ran an additional analysis using a compound dirichlet prior for branch lengths. The results are only minimally different compared to the analysis using the exponential distribution. In particular the position of the fossils described in our paper has remained unchanged. Considering the issues raised by Rannala et al. 2021 and Zhang et al. 2012, however, we decided to show only the new analysis.

I am also curious as to why you've chosen to perform a maximum likelihood analysis, instead of a parsimony analysis, as ML analyses of morphological data using the MK model have been shown to underperform parsimony analyses in terms of accuracy (Puttick et al. 2017, below). I would suggest you also run a parsimony analysis, in addition to your model-based approaches.

The aim of running another analysis in addition to the Bayesian one was to test if the position of the new fossil taxa is confirmed using other approaches. In doing so, we decided to opt for another model-based approach but given the literature on comparisons between various approaches, it is indeed more useful to add a parsimony-based approach. We therefore ran an additional analysis using parsimony and decided to present all three results (Bayesian, ML and parsimony).

Additional notes:

Page 5, line 24. You note that your ML tree produced a fully resolved tree. Its worth noting that all ML trees are always fully resolved, so you should consider presenting this result differently in the manuscript.

Fixed.

Reviewer 1:

p. 4 l 23: edged should be edges

Corrected.

p. 4 ls 44-45: delete Rambaut and URL, as you mentioned that already above

Deleted.

p. 11 l. 29/30: "belong" should be "belongs"

Corrected.

Table 1: should be ...matrix "of" Thuy and Stöhr., because the species were not added by Thuy and Stöhr

Corrected.

Fig. 2 caption: I think it should be "phylogeographic" instead of "phylogeographical"

Corrected (N.B.: palaeogeographic not phylogeographic)

In the Bayesian tree (fig. 3), a few numbers seem to overlap.

Fixed.